# Kinematic alteration in three-dimensional reaching movement in C3-4 level cervical myelopathy

Naoto Noguchi[1], Ryoto Akiyama[1�], Ken Kondo[2�], Duy Quoc Vo[3‡], Lisa Sato[4], Akihito Yanai[5‡], Masatake Ino[6], Bumsuk Lee[1]*

1 Gunma University Graduate School of Health Sciences, Maebashi, Gunma, Japan, 2 Department of Occupational Therapy Faculty of Rehabilitation, Gunma Paz University, Takasaki, Gunma, Japan, 3 Gunma University Graduate School of Health Sciences Doctoral Program, Maebashi, Gunma, Japan, 4 Department of Rehabilitation, Harunaso Hospital, Takasaki, Gunma, Japan, 5 Non-Profit Organization Sonrisa, Maebashi, Gunma, Japan, 6 Gunma Spine Center, Harunaso Hospital, Takasaki, Gunma, Japan

l These authors contributed equally to this work.
‡ These authors also contributed equally to this work
* leebumsuk@gunma-u.ac.jp

**Data Availability Statement:** All relevant data are within the paper and its Supporting Information files.

## Abstract

### Object

This study aimed to compare the reaching movement between two different spinal cord compression level groups in cervical myelopathy (CM) patients.

### Methods

Nine CM patients with maximal cord compression at the C3-4 level (C3-4 group) and 15 CM patients with maximal cord compression at the C4-7 level (C4-7 group) participated in the study. We monitored three-dimensional (3D) reaching movement using an electronic-mechanical whack-a-mole-type task pre-and post-operatively. Movement time (MT) and 3D movement distance (MD) during the task were recorded. An analysis of variance for split-plot factorial design was performed to investigate the effects of compression level or surgery on MT and MD. Moreover, we investigated the relationship between these kinematic reaching parameters and conventional clinical tests.

### Results

The 3D reaching trajectories of the C3-4 group was unstable with higher variability. The C3-4 group showed longer MT ($p < 0.05$) and MD ($p < 0.01$) compared with the C4-7 group both before and after surgery. Moreover, MT was negatively correlated with the Japanese Orthopedic Association score only in the C3-4 group (r = - 0.48).

### Conclusion

We found that spinal cord compression at the C3-4 level had a negative effect on 3D reaching movement and the kinematic alteration influenced the upper extremity performance. This new knowledge may increase our understanding of kinematic alteration in patients with CM.

**Funding:** Our study was supported by JSPS KAKENHI under Grant number 21K17507. Moreover, the funders had no role in our study design, data collection and analysis, decision to publish, or preparation of the manuscript.

**Competing interests:** The authors have declared that no competing interests exist.

## Introduction

Cervical myelopathy (CM) is a disorder caused by spinal cord compression associated with cervical spondylosis, disc herniation, and congenital stenosis [1]. CM patients generally suffer from sensory and motor symptoms such as paresthesia, muscle weakness, and clumsiness [2]. These impairments result in a decline in activities of daily living (ADLs) and quality of life [3]. In severe cases, cervical decompression surgery is necessary to relieve compression of the spinal cord.

Conventional clinical tests have been mainly focused on the distal arm dysfunction in CM. For example, the 10-s grip and release (G&R) test, which is one of the most commonly used clinical tests, is focused on the motor coordination between finger flexion and extension [4]. Similarly, CM symptoms tend to be characterized by impairments in hand dexterity such as loss of sensitivity, exaggerated deep tendon reflexes, and motor weakness [4,5]. However, pathophysiological studies on spinal cord compression identified that appearance of upper extremity symptoms depends on the level of spinal cord compression. Patients with C3-4 compression present proximal and distal arm weakness and sensory disturbances, whereas only distal arm dysfunction in patients with C6-7 level [6,7].

Previous studies suggested several possible reasons why proximal dysfunction in CM has not been adequately discussed compared to impairments in hand dexterity. First, the lower cervical spinal cord level is commonly affected in CM. Northover et al [8] reported that C5-6 level was the most severely and commonly affected level in patients, implying that distal arm dysfunction tends to occur more frequently compared to proximal dysfunction. Second, in multi-level cord compression distal arm dysfunction is easily perceived by the patients and clinicians [9]. Lastly, it is known that there are interneuronal systems which compensate for proximal arm dysfunction. In healthy individuals, propriospinal neurons (PN) at C3-4 level take the role of mediating corticomotoneuronal inputs and transmits proprioceptive feedback to control arm position during target reaching [10,11]. When damage occurs at C3-4 level, these feedforward and feedback control systems play a critical role in compensating decreased proximal arm function [12].

In addition to these pathophysiological aspects, we suppose that clinical tests for upper extremity function have inherent limitations in the assessment of the proximal arm function of CM patients. Clinical tests generally evaluate the reaching, grasping, and manipulation capability simultaneously, therefore it is difficult to separate proximal function from distal function. A fact worthy to note is that a discrepancy in reaching and grasping functions can be occurred in patients with CM. In order to solve this problem, several studies used an accelerometer to independently measure reaching function [13,14]. Although the sensor makes it possible to easily measure kinematic changes during reaching movement, data obtained from only one-axis acceleration (e.g., forward-backward direction) is still not enough to understand upper limb movement in three-dimensional (3D) space [13]. Recently, several techniques have been developed for synchronizing multiple accelerometers or combining three-axis acceleration data [14], but the necessity of a special calculation program and the time cost of data analysis limit its practical use in clinical environments. Therefore, the evaluation not depending on grasping or manipulation and detecting the changes in 3D reaching movements is necessary to accurately measure proximal arm function in CM.

The purpose of the study was to examine the sole reaching function in 3D space without being affected by distal function. In order to separate proximal from distal function, we used an electronic-whack-a-mole-type task which only requires shoulder and elbow joint movements, not accompanied by forearm or finger movements. We monitored the reaching movement time and distance during the task, suggesting that these parameters can detect the changes in any direction of reaching movement. We hypothesized that patients with C3-4 compression may have decreased reaching capability and the kinematic alteration has a negative influence on the upper extremity performance in CM patients.

## Materials and methods

### Participants

This prospective cross-sectional study was conducted from April 7, 2018 to November 29, 2020. Twenty-four CM patients who were waiting for cervical decompression surgery were recruited. Inclusion criteria were as follows: (1) performing pre-operative Magnetic resonance imaging (MRI), (2) having no visual disturbances or cognitive decline, (3) having no history of previous cervical spine surgery or other orthopedic disorders that may impair reaching movement, and (4) ability to walk with/without a walking aid. Participants were diagnosed with cervical spondylotic myelopathy (n = 22), ossification of the posterior longitudinal ligament (n = 1), or cervical spondylotic radiculopathy (n = 1). Handedness was evaluated with the Edinburgh Handedness Inventory, and all participants were right-handed.

The participants were classified into two subgroups, depending on the segmental level of maximal cord compression: a C3-4 group (maximal cord compression at C3-4 level, five men and four women, 74.3 ± 9.4 years), and a C4-7 group (maximal cord compression at C4-7 level, seven men and eight women, 65.2 ± 12.8 years). They underwent decompression surgery by laminoplasty (n = 17), laminectomy (n = 4), anterior spinal fusion (n = 2), or posterior spinal fusion (n = 1). Data from nine participants with lumbar stenosis (six men and three women, 62.0 ± 11.1 years) were collected for controls. Informed consent was obtained from all participants before the study. This study was approved by the Hospital Institutional Review Board (Approval number: 180101) and the University Ethical Review Board for Medical Research Involving Human Subjects (Approval numbers: HS2018-267 and HS2020-072).

### Electronic-mechanical whack-a-mole-type task

Participants performed a task by standing in front of a height-adjustable table. An electronic-mechanical whack-a-mole-type task (K3000, Work Joy, Japan, 31.8 × 23.5 cm) was positioned at a 45-degree angle and 20 cm away from the participants (Fig 1A). Verbal instructions were given to tap the start icon with a wooden hand-held pen (1.5 cm in diameter and 21.5 cm long) and reach for 16 targets (3.5 cm in diameter) that appeared one by one on the screen (Fig 1B). The first target appeared on the screen when participants tapped the start icon, and the following target simultaneously appeared when the previous target was touched. The distance between the two paired targets ranged from 19.0 to 25.5 cm. Each participant performed two trials, and a 5-min rest was provided between trials. CM groups performed the task pre- (2.6 ± 1.8 days prior to operation) and post-operatively (18.0 ± 8.5 days after the operation), and the control group performed the task only postoperatively (35.2 ± 23.7 days after the operation).

During the task, the position of the pen tip in 3D space was monitored using a Leap Motion Controller (LMC) with a frequency of 128 Hz in a spatial resolution of 0.7 mm [15]. The LMC consisted of three light-emitting diodes (LEDs) and two stereo cameras attached to the bottom of the screen. The cameras captured the reflection of the light, which the LEDs irradiated to

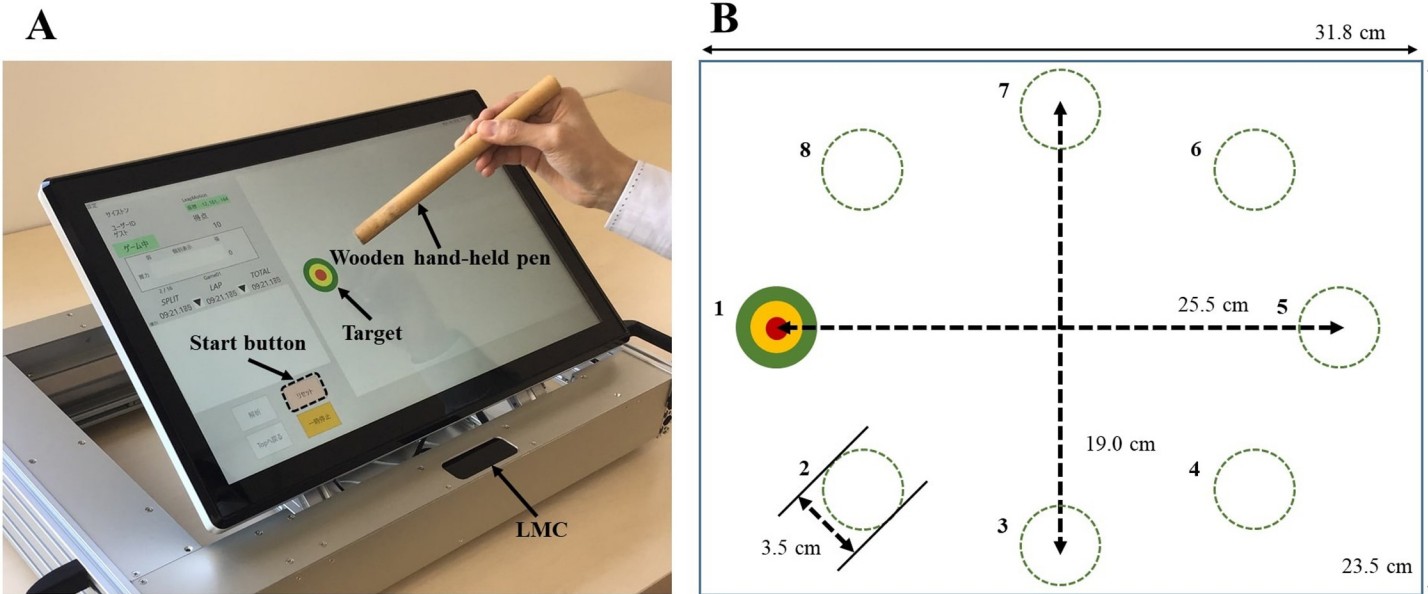

**Fig 1. Experimental apparatus.** (A) Electronic-mechanical whack-a-mole-type task: Participants use the wooden hand-held pen to reach and touch targets on the screen as accurately and quickly as possible. A Leap Motion Controller (LMC) was attached to the bottom of the screen and captured the three-dimensional position of the pen tip with a frequency of 128 Hz. (B) The sequence of target appearance: For example, when a participant touches "1", a symmetrically located target ("5") appears on the screen simultaneously. After touching "5", another target appears randomly. If a "3" appears, the next target is always "7", which is symmetrically located on the screen.

the pen in an inverted pyramid 3D space (60 × 60 × 60 cm) during the task. Previous studies reported the reliability and validity of computer-stimulated movement tracking system using LMC for evaluating hand dexterity dysfunction in CM [16,17]. Based on the spatial and temporal information of the pen tip, we calculated two kinematic parameters as follows: (1) movement time (MT): the mean movement time of reaching between paired targets, and (2) 3D movement distance (MD): the mean trajectory distance of the pen tip between paired targets. In our previous study, the intra-subject repeatability of these parameters was assessed using intraclass correlation coefficients (ICC), and the parameters demonstrated substantial reliability (MT: ICC $_{(1, 1)}$ = 0.83 and MD: ICC $_{(1, 1)}$ = 0.60) [18].

## Clinical tests

The latest version of the Japanese Orthopedic Association (JOA) score was used. The score consists of six domains: motor function in the upper extremities, motor function in the lower extremities, sensory function in the upper extremities, sensory function in the trunk, sensory function in the lower extremities, and bladder function. The minimum total score is 0 and the maximum is 17. In the present study, we only used the subscore of the upper extremities based on previous studies [2,19], and defined it as JOA-UEF [20]. The JOA-UEF was assessed from 0 to 4 points: 0 = unable to feed oneself with any tableware, including chopsticks, a spoon, or fork, and/or unable to fasten buttons of any size, 1 = can manage to feed oneself with a spoon and/or a fork but not with chopsticks, 2 = either chopstick feeding or writing is possible but not practical, and/or large buttons can be fastened, 3 = either chopstick feeding or writing is clumsy but practical, and/or cuff buttons can be fastened, and 4 = normal.

The maximum grip strength was measured with the Smedley Grip Tester (Gopher, Owatonna, MN, USA), and pulp pinch strength between thumb and index finger was measured with the JAMAR Hydraulic Pinch Gauge (Petterson Medical, Warrenville, IL, USA). The

cutaneous pressure threshold of the central area of the thumb, index, and middle finger pulps were assessed with Touch Test Sensory Evaluators (North Coast Medical Inc., Morgan Hill, CA, USA), which apply standard calibrated forces of 0.07, 0.4, 2.0, 40, and 300 g. The lowest perceived monofilament is considered normal, and higher monofilament forces indicate severe sensory disturbance. In the 10-s G&R test, participants were asked to open and close their hands as quickly and fully as possible in 10 seconds. Hand dexterity function was evaluated by three subtests time (numbers 8, 9, and 10) of the Simple Test for Evaluating Hand Function (STEF) [21]. Total time of the three subtests was used for statistical analysis.

## Statistical analysis

A one-way analysis of variance (ANOVA) was performed to compare demographic data and kinematic parameters among three groups. Tukey's test was applied as a post hoc test to identify specific differences.

Secondly, a split-plot ANOVA was performed to investigate differences within CM groups. The factors were 'compression level' with two levels (maximal cord compression at C3-4 level and C4-7 level, and 'surgery' with two levels (pre- and post-operation).

Associations of the reaching parameters and other clinical tests with the JOA-UEF were examined in each CM group. Statistical analyses were performed using statistical software SPSS ver. 27.0 J for Windows (SPSS Japan., Tokyo, Japan). Values of $p < 0.05$ were considered significant.

## Results

Fig 2 shows the 3D reaching trajectories during the task for representative patients. The control group patient moved between the targets in parabolic path with lower variability (Fig 2A). In contrast, the trajectory of the patient in C3-4 group was unstable with higher variability (Fig 2B).

Table 1 summarizes the demographic data and kinematic outcomes for the participants. Age, sex, body mass index, and disease duration were not different among the three groups. On the other hand, the one-way ANOVA and Tukey post-hoc analysis indicated that the C3-4 group required significantly longer MT (vs. control group; $p < 0.05$) and longer MD (vs. C4-7 group; $p < 0.01$).

Table 2 shows the results of the comparisons within CM groups. A split-plot ANOVA revealed the significant main effect of compression level in MT ($p = 0.02$), MD ($p < 0.01$), and JOA-UEF score ($p = 0.04$), indicating deficits in reaching function and the upper extremity performance in the C3-4 group. The main effect of surgery was found in the JOA-UEF score ($p < 0.01$) and the G&R test ($p < 0.01$), suggesting that upper extremity performance and finger coordination improved postoperatively in both CM groups. A synergistic interaction was found in the JOA-UEF score ($p < 0.01$).

The results of the correlations between the JOA-UEF score and the other tests are shown in Table 3. In the C3-4 group, the JOA-UEF score was associated with MT ($r = -0.48$, $p = 0.04$), the G&R test ($r = 0.57$, $p = 0.02$), and the STEF time ($r = -0.57$, $p = 0.01$). In the C4-7 group, the JOA-UEF score was correlated with the cutaneous pressure threshold of the fingers (thumb: $r = -0.41$, $p = 0.02$; index: $r = -0.43$, $p = 0.02$; middle: $r = -0.48$, $p < 0.01$), and the G&R test ($r = 0.52$, $p < 0.01$).

## Discussion

We found that spinal cord compression at the C3-4 level had a negative effect on reaching movement. Moreover, the correlation between MT and the JOA-UEF was only found in the

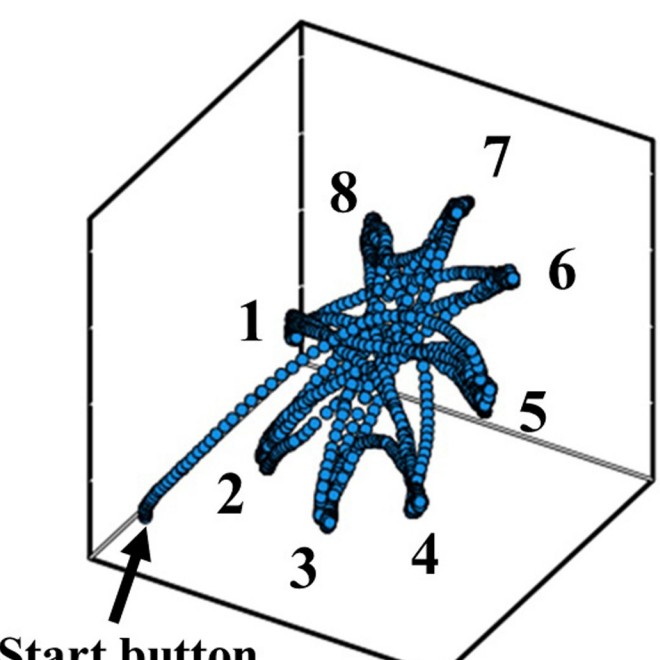 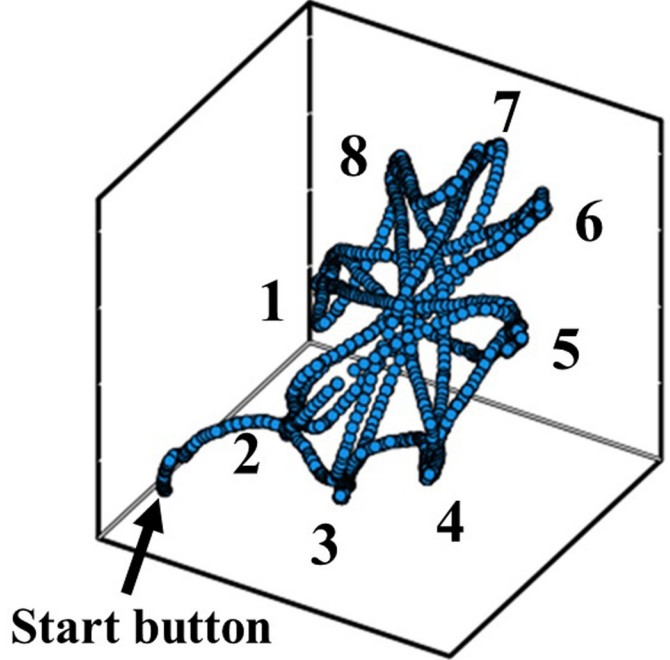

**Fig 2. The three-dimensional reaching trajectories during the task for representative patients.** The dots indicate the spatial position of the pen tip every 7.8ms, and the numbers indicate the location of targets on the screen. The control group patient moves between the targets in parabolic path with lower variability (Fig 2A). In contrast, the trajectory of the patient in C3-4 group was unstable with higher variability (Fig 2B).

**Table 1. Demographic data and kinematic outcomes based on one-way ANOVA.**

|  | C3-4 (n = 9) | C4-7 (n = 15) | Control (n = 9) |
|---|---|---|---|
| Demographic |  |  |  |
| Age (years) | 74.3 ± 9.4 | 65.2 ± 12.8 | 62.0 ± 11.1 |
| Sex (male/female) | 5 / 4 | 7 / 8 | 6 / 3 |
| BMI (kg/m$^2$) | 26.3 ± 3.5 | 23.4 ± 3.4 | 25.4 ± 4.4 |
| Disease duration (months) | 27.4 ± 53.7 | 27.9 ± 25.2 | 30.9 ± 23.7 |
| Reaching movement |  |  |  |
| Movement time (s) [a] | 0.83 ± 0.13 | 0.69 ± 0.17 | 0.63 ± 0.06 |
| Movement distance (m) [b] | 0.27 ± 0.03 | 0.24 ± 0.02 | 0.26 ± 0.03 |

BMI, body mass index.

Tukey's test.

[a] Significant difference between C3-4 and control groups ($p < 0.05$).

[b] Significant difference between C3-4 and C4-7 groups ($p < 0.01$).

**Table 2. Effects of compression level or surgery based on ANOVA for split-plot factorial design.**

| | C3-4 (n = 9) | | C4-7 (n = 15) | | Compression level effect | | Surgical effect | | Interaction | |
|---|---|---|---|---|---|---|---|---|---|---|
| | Pre-ope | Post-ope | Pre-ope | Post-ope | F value | P value | F value | P value | F value | P value |
| Reaching movement | | | | | | | | | | |
| Movement time (s) | 0.83 ± 0.13 | 0.75 ± 0.13 | 0.69 ± 0.17 | 0.65 ± 0.10 | 5.84 | **0.02** | 4.06 | 0.06 | 0.54 | 0.47 |
| Movement distance (m) | 0.27 ± 0.03 | 0.26 ± 0.03 | 0.24 ± 0.02 | 0.24 ± 0.01 | 12.29 | **< 0.01** | 1.00 | 0.33 | 1.93 | 0.18 |
| JOA-UEF | 2.22 ± 0.97 | 4.00 ± 0.00 | 3.13 ± 0.74 | 3.93 ± 0.26 | 4.77 | **0.04** | 58.92 | **< 0.01** | 8.48 | **< 0.01** |
| Grip strength (kg) | 22.33 ± 10.31 | 23.78 ± 9.72 | 25.82 ± 10.50 | 24.20 ± 12.26 | 0.20 | 0.66 | < 0.01 | 0.95 | 1.18 | 0.29 |
| Pinch strength (kg) | 3.62 ± 1.69 | 3.62 ± 1.34 | 4.06 ± 1.87 | 3.79 ± 1.36 | 0.22 | 0.64 | 0.56 | 0.46 | 0.61 | 0.44 |
| Cutaneous pressure threshold (g) | | | | | | | | | | |
| Thumb | 5.30 ± 13.04 | 0.43 ± 0.61 | 23.09 ± 77.28 | 0.18 ± 0.16 | 0.45 | 0.51 | 1.12 | 0.30 | 0.47 | 0.50 |
| Index finger | 5.08 ± 13.12 | 0.39 ± 0.62 | 20.41 ± 77.35 | 0.22 ± 0.50 | 0.33 | 0.57 | 0.90 | 0.35 | 0.35 | 0.56 |
| Middle finger | 5.22 ± 13.07 | 0.22 ± 0.17 | 23.05 ± 77.29 | 0.35 ± 0.68 | 0.47 | 0.50 | 1.12 | 0.30 | 0.46 | 0.51 |
| 10-s G&R test (number) | 15.44 ± 6.82 | 19.67 ± 6.25 | 18.29 ± 5.25 | 22.14 ± 6.16 | 1.34 | 0.26 | 11.75 | **< 0.01** | 0.02 | 0.88 |
| STEF (s) | 48.23 ± 18.51 | 40.05 ± 5.11 | 39.66 ± 12.31 | 41.83 ± 28.00 | 0.27 | 0.61 | 0.37 | 0.55 | 1.09 | 0.31 |

JOA, Japanese Orthopedic Association; UEF, upper extremity function; G&R, grip and release; STEF, Simple Test for Evaluating Hand Function.

Values in bold indicate significant difference.

C3-4 group, implying cord compression at the C3-4 level has a critical influence on the upper extremity performance.

 The key finding in the present study is that the alteration of reaching movement was found in C3-4 level myelopathy (Tables 1 and 2). Although reaching movement in CM patients has been assessed, the influence of compression level was not adequately discussed. To the best of our knowledge, our study is the first report to reveal that the alteration of reaching movement may differ depending on the compression level. This finding can be explained by the methodological uniqueness of our experimental technique. It is known that the inability to extend fingers and grip/release rapidly is one of the most common hand dysfunction in CM patients [4].

**Table 3. Correlations between the reaching movement and upper extremity functional tests in each cervical myelopathy group.**

| | JOA-UEF | | | |
|---|---|---|---|---|
| | C3-4 | | C4-7 | |
| | r | P value | r | P value |
| Reaching movement | | | | |
| Movement time (s) | - 0.48 | **0.04** | - 0.23 | 0.23 |
| Movement distance (m) | - 0.45 | 0.06 | 0.16 | 0.39 |
| Grip strength (kg) | 0.27 | 0.27 | - 0.10 | 0.60 |
| Pinch strength (kg) | 0.13 | 0.61 | 0.10 | 0.59 |
| Cutaneous pressure threshold (g) | | | | |
| Thumb | - 0.28 | 0.26 | - 0.41 | **0.02** |
| Index finger | - 0.28 | 0.26 | - 0.43 | **0.02** |
| Middle finger | - 0.29 | 0.25 | - 0.48 | **< 0.01** |
| 10-s G&R test (number) | 0.57 | **0.02** | 0.52 | **< 0.01** |
| STEF (s) | - 0.57 | **0.01** | - 0.10 | 0.62 |

n = 48 (24 individuals × 2 time points).

Values in bold indicate significant correlation.

However, previous studies used pointing movement to a target with the extended index finger [12] or reach-to-grasp movement [2]. We hypothesized reaching movement can be affected by distal function, and introduced the technique not accompanying the movement of distal joints. As a result, the alteration of reaching movement in C3-4 level myelopathy was found. Thus, it is reasonable to assume that separation of proximal function from distal is critical to know the influence of compression level on proximal upper extremity function in CM patients.

Another reason for the finding of the compression level-dependent difference could be related to the higher level of task difficulty in our experiment. We assume that single-directional reaching movement in a horizontal plane, which has been typically used in many studies in CM [2,12] is too easy to identify compression level-dependent difference. Target direction and height of reaching movement affect reaching performance. For example, MT in the lateral target is significantly longer than in the medial target, even though the distance is equal [22]. Endpoint error increases when reaching a high place compared to low-reaching movement [23]. In addition, the unpredictability of target positions also can be related to the difficulty of the task. During the whack-a-mole-type task in our experiment, the following target simultaneously and pseudo-randomly appeared when the previous target was touched. This does not allow our participants to prepare a precise control of target reaching. The influence is consistent with a previous study that reported increased MT and MD when reaching a target shifting unexpectedly [24]. Taken together, we suggest that the high level task incorporating unpredictable multidirectional reaching conditions could be useful for obtaining more detailed information on the proximal upper limb dysfunction in CM.

In C3-4 group, the JOA-UEF was associated with distal motor function, but not with distal sensory function (Table 3). Before the experiment, based on the idea that CM symptoms typically occur at and below the compression level, we expected that the JOA-UEF may correlate with both proximal and distal arm sensorimotor function in C3-4 group. As expected, MT and G&R test were associated with the JOA-UEF, implying the critical relationship between proximal/distal limb motor function and daily performance. On the other hand, interestingly, distal sensory function was not related with daily performance. The discrepancy between distal motor and sensory functions in C3-4 group could be explained by a pathophysiological model of CM. The model suggests that CM symptom progression can be divided into three subgroups depending on the progress of myelopathy [25]. First, gray matter in the central section of spinal cord is damaged, and motor and sensory dysfunction occur at the compression level (Type ). Then, pyramidal tract lesion occurs, and as a result motor dysfunction appears below the compression level is added (Type ). Lastly, the lesion extends to spinothalamic pathway, which leads below level sensory dysfunction (Type ). When we interpret our finding based on this pathophysiological classification, it is reasonable to assume that the influence of motor and sensory functions on daily performance can be changed in accordance with the progress in CM.

Although both the JOA-UEF and the STEF evaluate upper limb performance in CM, the surgical effect was only found in the JOA-UEF score (Table 2). This implies that the upper extremity function was improved only in the practical use in eating or dressing, but not in the ability to rapidly carry objects. One potential reason may be a sequence of stepwise recovery in the motor function. In patients with stroke, for example, the appearance of voluntary movement always precedes the ability to produce fast moments [26]. They can first grasp and release objects slowly with their paretic hand. Movements that require speed are always obtained in the last recovery stage. A similar stepwise sequence can be found in tendon injury or burn in the hand and upper extremity [27,28]. Indeed, a previous CM study reported that the JOA-UEF score improved 1 week after surgery, on the other hand, the postoperative improvement in the STEF required 1 month [19]. Monitoring both practical use in daily living and

maximal capacity is, therefore, necessary for understanding the longitudinal recovery process in upper extremity performance in CM.

Several limitations of the present study should be acknowledged. Firstly, we only analyzed the total time and distance required reaching movement between targets. It was reported that multiple peak velocities are detected in a single reaching movement in patients with sensori-motor impairment [29]. Additional kinetic parameters analyzing more detailed temporal/spatial changes need be considered to identify which phase during target reaching was affected by spinal cord compression. Secondly, we did not consider a capability of static standing balance during the reaching task. It would be possible that the postural adjustment was an uncontrolled factor in this study. Simultaneous measurement with displacement trajectory using a force platform may clarify the dynamic interaction between the reaching and balance strategies.

In conclusion, this study demonstrated that spinal cord compression at the C3-4 level had a negative effect on reaching movement, and alteration in reaching movement was correlated with the severity of upper extremity performance. This new knowledge may increase our understanding of kinematic alteration in CM patients.

## Supporting information

**S1 File. Data set.**
(XLSX)

## Author Contributions

**Conceptualization:** Naoto Noguchi.

**Data curation:** Naoto Noguchi.

**Formal analysis:** Naoto Noguchi, Ryoto Akiyama.

**Funding acquisition:** Naoto Noguchi, Ken Kondo.

**Investigation:** Naoto Noguchi, Lisa Sato, Masatake Ino.

**Project administration:** Bumsuk Lee.

**Supervision:** Bumsuk Lee.

**Writing – original draft:** Naoto Noguchi.

**Writing – review & editing:** Ryoto Akiyama, Ken Kondo, Duy Quoc Vo, Lisa Sato, Akihito Yanai, Bumsuk Lee.

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
