## [Decision Letter · Decision Letter 0]

18 Oct 2023

PONE-D-23-26285Kinematic alteration in three-dimensional reaching movement in C3-4 level cervical myelopathyPLOS ONE

Dear Dr. Lee,

Thank you for submitting your manuscript to PLOS ONE. After careful consideration, we feel that it has merit but does not fully meet PLOS ONE’s publication criteria as it currently stands. Therefore, we invite you to submit a revised version of the manuscript that addresses the points raised during the review process.

ACADEMIC EDITOR: 

The decision for the manuscript entitled "Kinematic alteration in three-dimensional reaching movement in C3-4 level cervical myelopathy" PONE-D-23-26285, is to perform minor adjustments as requested by one of the reviewers. Pleas find the comment below ( in section 5 Review Comments to the Author).

Please submit your revised manuscript by Dec 02 2023 11:59PM. If you will need more time than this to complete your revisions, please reply to this message or contact the journal office at plosone@plos.org. Please include the following items when submitting your revised manuscript:A rebuttal letter that responds to each point raised by the academic editor and reviewer(s). You should upload this letter as a separate file labeled 'Response to Reviewers'.A marked-up copy of your manuscript that highlights changes made to the original version. You should upload this as a separate file labeled 'Revised Manuscript with Track Changes'.An unmarked version of your revised paper without tracked changes. You should upload this as a separate file labeled 'Manuscript'.If applicable, we recommend that you deposit your laboratory protocols in protocols.io to enhance the reproducibility of your results. Protocols.io assigns your protocol its own identifier (DOI) so that it can be cited independently in the future. For instructions see: https://journals.plos.org/plosone/s/submission-guidelines#loc-laboratory-protocols. Additionally, PLOS ONE offers an option for publishing peer-reviewed Lab Protocol articles, which describe protocols hosted on protocols.io. Read more information on sharing protocols at https://plos.org/protocols?utm_medium=editorial-email&utm_source=authorletters&utm_campaign=protocols.

We look forward to receiving your revised manuscript.

Kind regards,

Nadinne Alexandra Roman, Ph.D.

Academic Editor

PLOS ONE

6. We note that Figure 1 in your submission contain copyrighted images. All PLOS content is published under the Creative Commons Attribution License (CC BY 4.0), which means that the manuscript, images, and Supporting Information files will be freely available online, and any third party is permitted to access, download, copy, distribute, and use these materials in any way, even commercially, with proper attribution. For more information, see our copyright guidelines: http://journals.plos.org/plosone/s/licenses-and-copyright.

Reviewers' comments:

Reviewer's Responses to Questions

**Comments to the Author**

1. Is the manuscript technically sound, and do the data support the conclusions?

Reviewer #1: Yes

Reviewer #2: Yes

2. Has the statistical analysis been performed appropriately and rigorously? 

Reviewer #1: Yes

Reviewer #2: Yes

3. Have the authors made all data underlying the findings in their manuscript fully available?

Reviewer #1: Yes

Reviewer #2: Yes

4. Is the manuscript presented in an intelligible fashion and written in standard English?

Reviewer #1: Yes

Reviewer #2: Yes

5. Review Comments to the Author

Reviewer #1: The authors performed a study addressing “Kinematic alteration in three-dimensional reaching movement in C3-4 level cervical myelopathy”. However, there is one unclear area. Why is it appropriate to monitor 3D reaching movement and measure movement time (MT) and 3D movement distance (MD) during the task to have the objectives of this study achieved? Is it difficult to do so by other methods? Please describe your reasons in more detail.

Reviewer #2: The article title was so intresting to me , that made me to read all the article ,

You can move foreward and , do some future researches after this study to conplete your study on whole spine ,

Thanks

6. PLOS authors have the option to publish the peer review history of their article (what does this mean?). If published, this will include your full peer review and any attached files.

Reviewer #1: No

Reviewer #2: No

---

## [Author Response · Author response to Decision Letter 0]

26 Oct 2023

Thank you very much for your suggestions and comments. We found your suggestions and comments were very helpful in improving our manuscript. In a rebuttal letter, we summarized specific responses to all instructions from the Academic Editor and queries raised by the reviewers.

---

## [Editor Report · Decision Letter 1]

16 Nov 2023

Kinematic alteration in three-dimensional reaching movement in C3-4 level cervical myelopathy

PONE-D-23-26285R1

Dear Dr. Lee,

We’re pleased to inform you that your manuscript has been judged scientifically suitable for publication and will be formally accepted for publication once it meets all outstanding technical requirements.

Kind regards,

Nadinne Alexandra Roman, Ph.D.

Academic Editor

PLOS ONE

---

## [Editor Report · Acceptance letter]

20 Nov 2023

PONE-D-23-26285R1 

Kinematic alteration in three-dimensional reaching movement in C3-4 level cervical myelopathy 

Dear Dr. Lee:

I'm pleased to inform you that your manuscript has been deemed suitable for publication in PLOS ONE. Congratulations! Your manuscript is now with our production department. 

Kind regards, 

on behalf of

Dr. Nadinne Alexandra Roman 

Academic Editor

PLOS ONE